# Adipose Tissue Paracrine-, Autocrine-, and Matrix-Dependent Signaling during the Development and Progression of Obesity

**DOI:** 10.3390/cells12030407

**Published:** 2023-01-25

**Authors:** Elizabeth K. Johnston, Rosalyn D. Abbott

**Affiliations:** Department of Biomedical Engineering, Carnegie Mellon University, Pittsburgh, PA 15213, USA

**Keywords:** white adipose tissue, angiogenesis, paracrine signaling, inflammation, vasoregulation, extracellular matrix, adipokine, obesity, fibrosis, atherosclerosis

## Abstract

Obesity is an ever-increasing phenomenon, with 42% of Americans being considered obese (BMI ≥ 30) and 9.2% being considered morbidly obese (BMI ≥ 40) as of 2016. With obesity being characterized by an abundance of adipose tissue expansion, abnormal tissue remodeling is a typical consequence. Importantly, this pathological tissue expansion is associated with many alterations in the cellular populations and phenotypes within the tissue, lending to cellular, paracrine, mechanical, and metabolic alterations that have local and systemic effects, including diabetes and cardiovascular disease. In particular, vascular dynamics shift during the progression of obesity, providing signaling cues that drive metabolic dysfunction. In this review, paracrine-, autocrine-, and matrix-dependent signaling between adipocytes and endothelial cells is discussed in the context of the development and progression of obesity and its consequential diseases, including adipose fibrosis, diabetes, and cardiovascular disease.

## 1. Introduction

Obesity is a complex phenomenon involving the accumulation of excess body fat. While there are genetic, behavioral, environmental, and hormonal influences on the development and progression of obesity, the basis of the disease lies in the premise of caloric excess, with this surplus being stored as triacylglycerols within adipose depots [1]. The prevalence of obesity has been continuously increasing, with approximately 42% of Americans being considered obese (body mass index (BMI) ≥ 30) and 9.2% being considered morbidly obese (BMI ≥ 40) between 2017 and 2018 [1,2]. With obesity being characterized as a chronic low-grade inflammatory disease, there are many alterations in the cellular populations within the tissue, lending to cellular, paracrine, mechanical, and metabolic alterations that have local and systemic effects, including cardiovascular and metabolic diseases. In particular, vascular dynamics shift during the progression of obesity, providing signaling cues that drive metabolic dysfunction [3].

Adipose tissue has a high degree of plasticity, whereby it can accommodate normal fluctuations in the body’s energy intake and demand by regulating tissue size. Adipose tissue can expand either through adipocyte hypertrophy or hyperplasia, and it can shrink in a state of fasting or exercise in order to supply the necessary free fatty acids to other organs. This shrinkage occurs through a lipolytic mechanism, where triacylglycerols are broken down into free fatty acids and glycerol. Adipocyte hypertrophy, the growth of adipocyte cell size, occurs through lipogenesis; this converts free fatty acids into triacylglycerols, which are stored within the cell’s lipid vacuole. Adipocyte hyperplasia occurs through the recruitment and differentiation of adipose-derived stem cells and preadipocytes into new adipocytes, thus increasing the overall adipocyte number. A unique feature of adipose tissue is the coordination between angiogenesis and adipogenesis with adult stem cells [3,4,5]. By modulating the vascularity of the tissue, the overall quantity of fat mass can be regulated [6]. For example, Rupnick et al. showed that angiogenesis inhibitors induced endothelial cell apoptosis, which resulted in adipose tissue regression and weight loss in *ob/ob* mice [7].

As demonstrated in Figure 1, with progressive obesity, there is a reduction in the quantity of adipogenic stem cells and an impairment in adipogenesis, resulting in hypertrophic lipid accumulation [8,9]. As a result, the uncontrolled adipocyte expansion reduces the overall capillary density (rarefaction) within the tissue, limits the diffusion of nutrients and oxygen, and creates a hypoxic and proinflammatory environment [6,9,10,11,12]. Epidemiological studies have shown that subcutaneous adipocyte cell size is negatively correlated with insulin sensitivity [8,13,14,15]. Additionally, these hypoxic, hypertrophic adipocytes are more likely to become necrotic, contributing to a cycle of hypoxia, cell death, and inflammation [16]. With the influx of inflammatory cells, including macrophages, into obese adipose tissue, cells within this environment, including adipocytes and endothelial cells, start to behave aberrantly, communicating with neighboring cells and their environment through paracrine- and matrix-derived signals, including inflammatory cytokines, adipokines, and vasoregulators [17]. In this review, paracrine-, autocrine-, and matrix-dependent signaling between adipocytes and endothelial cells is discussed in the context of the development and progression of obesity and its consequential diseases.

## 2. Paracrine Signals Involved in Obesity

In response to sustained hypoxia due to hypertrophied adipocytes, resident macrophages undergo a phenotypic switch toward an M1 phenotype. There is then the further recruitment and infiltration of circulating monocytes that are then directed toward this proinflammatory phenotype [18]. These macrophages are notorious for their pernicious cytokines, including Interleukin-6 (IL-6) and Tumor Necrosis Factor-α (TNF). These potent molecules run rampant in obesity, having direct adverse effects on the adipocyte niche, as well as on the stromal vascular fraction, as summarized in Table 1. Additionally, these two cytokines are primary regulators for the release of other adipokines and vasoregulators, as noted in Figure 2, including adiponectin, leptin, nitric oxide (NO), and endothelin-1 (ET-1).

### 2.1. Inflammatory Cytokines

TNF is synthesized as a transmembrane protein that presents on the cell surface [19]. Upon cleavage by the TNF-converting enzyme, the soluble form of TNF is shed from the cell surface [19,20]. While transmembrane TNF is prevalent on obese adipocytes, the primary producers of soluble TNF are preadipocytes and macrophages within obese tissue [19]. Clinical studies have shown that individuals with obesity have increased levels of TNF-α in their sera and that these levels decrease with weight loss [21].

The implications of this chronic TNF elevation in adipose tissue during obesity have been studied in vitro through the addition of soluble TNF to the tissue’s cellular constituents, including adipocytes, preadipocytes, and endothelial cells. TNF has been shown to be a lipolytic stimulant of adipose cells cultured in vitro, including adipocytes and preadipocytes [22,23]. Additionally, Weiner et al. found that 3T3-L1-derived adipocytes dedifferentiated and subsequently upregulated transforming growth factor-β (TGF-β) secretion in the presence of 5 nM TNF [24]. This further resulted in the upregulation of extracellular matrix secretions, including collagen types I, III, and IV [24]. This transdifferentiation phenomenon was similarly seen when preadipocytes cultured with TNF-α experienced preferential differentiation toward a macrophage-like phenotype, as indicated by the expressions of Granulocyte-Macrophage Colony Stimulating Factor (GM-CSF), Interleukin-1β (IL-1β), Macrophage Inflammatory Protein-1α (MIP-1α), TNF-α, CD68, and matrix metalloprotease 3 (MMP3) [25,26,27].

The development of endothelial dysfunction has been reported to occur as a result of adipose-tissue-derived TNF. Upon incubation with obese adipocyte-derived infranatant, human umbilical vein endothelial cells (HUVECs) significantly upregulated their VCAM-1, ICAM-1, and E-selectin receptors in a TNF-dependent manner [28]. Not only do these endothelial cells become more receptive to inflammatory cells in response to TNF, but they also begin to have a more proinflammatory secretome, as signified by the secretion of IL-8, MIP-1β, MIP-3α, Monocyte Chemoattractant Protein-1 (MCP-1), and IL-6 [29,30]. Finally, Haynes et al. identified the occurrence of endothelial-to-mesenchymal transition in obese adipose tissue, as indicated by the colocalization of CD31 and α-SMA in the capillaries of obese adipose tissue [31]. This phenomenon was also seen when healthy adipose-tissue-derived endothelial cells were exposed in vitro to either TNF, TGF-β, or IFN-γ. These endothelial cells exposed to inflammatory signals underwent this transition to obtain a mesenchymal-like phenotype, where they were unable to maintain tight barriers in the culture, had enhanced migration and reduced angiogenesis, released proinflammatory extracellular vesicles, and experienced reductions in oxidative phosphorylation and glycolysis [31]. It is apparent that there is potential for the inflammatory signal TNF to induce both local and systemic effects.

However, the role of TNF in the initiation of insulin resistance remains unclear. While TNF secretion has been speculated to be involved in the initiation of insulin resistance [32], much of the research has been performed in high-fat diet-fed animals [33] or with differentiated murine cells lacking cellular interactions [34], both of which do not mimic human physiology or pathophysiology. Human adipocytes in co-culture with stromal vascular cells (endothelial cells, stem cells, and pericytes [35]) responded to TNF stimulation with an increase in glucose uptake [36]. Despite the positive correlations found between plasma TNF levels and peripheral insulin resistance, the neutralization and antagonization of TNF have not been shown to improve insulin resistance in humans [37,38,39].

While TNF-α acts locally, adipose-derived interleukins have metabolic and inflammatory effects both locally and systemically [40,41]. Interleukins are cytokines largely secreted by leukocytes, but, more recently, they have been found to both be produced by and act upon adipose-specific cells, including adipose stem and progenitor cells [42]. Composed of close to 50 different interleukins, this family of cytokines plays a crucial role in regulating both the innate and adaptive immune systems, as well as in both metabolic and regenerative processes [40,43]. For more information regarding the breakdown of these adipose-relevant interleukins, please refer to the following review articles: [43,44].

It is well-documented that IL-6 secretion is upregulated in obesity, with positive relationships being found between adipose tissue mass, adipocyte cell size, and IL-6 secretion [45,46,47,48,49]. This proinflammatory cytokine induces effects on adipocytes similar to those of TNF, including lipolytic stimulation and leptin secretion [22,46,50]. Importantly, this cytokine has been found to be strongly positively correlated with the amount of non-esterified free fatty acids in the serum and negatively correlated with whole-body insulin sensitivity, showing the detrimental effects of this cytokine [45].

Many cells, including adipocytes, macrophages, and endothelial cells, both secrete and respond to IL-6, giving them autocrine, paracrine, and endocrine effects [46,51,52]. However, the cellular origin of the cytokine determines the effect that it has within adipose tissue. Han et al. used IL-6 knockout mice to study the source-specific role of IL-6 in the development of obesity. Here, it was found that adipocyte-derived IL-6 promotes high-fat diet-induced adipose tissue inflammation, while myeloid-derived, adipocyte-targeted IL-6 prevents adipose tissue macrophage accumulation and improves glucose and insulin tolerance through the canonical pathway [48]. Further, it was found that these differential responses are determined by the signaling pathway that they go through. The classical signaling pathway is conducted through the membrane-bound IL-6 receptor (IL-6R) and is characterized by PI3K-Akt and ERK ½ activation, resulting in anti-inflammatory responses [43,48,53]. It is also possible for signaling to go through the trans-signaling pathway, which depends on the ADAM 10/17-cleaved, soluble IL-6 receptor, which then activates the STAT3 pathway if the soluble form of the IL-6 receptor is present [54]. This IL-6 *trans*-signaling pathway has proinflammatory results [43,48,53].

Cytokines IL-1, TNF, and IFN-γ have been shown to stimulate IL-6 release from HUVECs [55]. However, this stimulated IL-6 did not have an autocrine effect despite the known presence of an IL-6R on endothelial cells [54,55]. By culturing endothelial cells with or without IL-6 or IL-6+soluble IL-6R, Montgomery et al. was able to observe the differential effects of the classical signaling and trans-signaling in endothelial cells [54]. The activation of the classical pathway in endothelial cells results in the inhibition of cell death induced via serum deprivation, but the activation of the trans-signaling pathway encourages the secretion of proinflammatory cytokines, such as MCP-1, and the upregulation of monocyte adhesion molecules on endothelial cells [54].

**Table 1 cells-12-00407-t001:** Prominent signaling molecules in adipose tissue.

Signaling Molecule		Physiological Role in Adipose Tissue	Obese State	Regulation	Citation
**Inflammatory** **Cytokines**	**TNF**	Involved in immunity Regulates the functions of immune cells, but is found in low levels	Increase in TNF in obesity **Stimulates:** Adipocyte lipolysis Leptin production IL-6 secretion Plasminogen Activator Inhibitor-1 (PAI-1) biosynthesis ROS production Activates NFκB Inhibits adipocyte differentiation	Triggered through adipocyte cell death	[10,19,25,46,56,57]
	**Interleukins**	Classical signaling through the IL-6 canonical pathway is anti-inflammatory and regulates glucose and insulin sensitivity IL-10 is a more prominent interleukin in lean adipose tissue	IL-1β, IL-6, and IL-8 increase adipose inflammation during obesity IL-6 induces lipolytic stimulation and leptin secretion from adipocytes through the trans-signaling pathway while upregulating MCP-1 secretion from endothelial cells that then upregulate their expression of monocyte adhesion molecules	Cellular source of IL-6 production regulates response Endothelial cells release IL-6 in response to TNF, IFN-γ, and IL-1	[10,46,54,58,59,60]
	**Interferons**	A balance in pro and anti-inflammatory interferons is maintained in order to properly identify and combat invading viruses. This is done through communication between the immune system and the target cells.	Type 1 interferons are upregulated through obesity-derived metabolic endotoxemia. Lipopolysaccharide (LPS) stimulates the production of IFN-β in mouse adipocytes Type II interferon, IFN-γ, is upregulated in obesity due to the influx of IFN-γ-secreting T cells in obese adipose tissue Promotes both a local and systemic inflammatory network between immune cells, adipocytes, and endothelial cells	Adiponectin limits IFN-γ release from CD4+ T cells	[61,62,63,64]
	**MCP-1**	Chemokine that is secreted by both immune and non-immune cells to attract monocytes/macrophages into the tissue	Increased circulating levels in individuals with obesity Adipocyte progenitors secrete MCP-1 to promote M1 macrophage accumulation in adipose tissue Induces endothelial apoptosis in vitro, thus promoting atherosclerosis	Secreted upon injury or ROS exposure Regulated by TNF and *trans*-IL-6 signaling Insulin-responsive gene	[25,27,54,65,66,67,68,69,70]
**Adipose Hormones**	**Leptin**	Produced by adipose tissue and the stomach but acts on the central nervous system (hypothalamus) in order to regulate satiety.	Increase in leptin production alongside an increase in fat mass Chronic hyperleptinemia can result in leptin resistance and atherogenic consequences Induces SPARC and collagen II and IV expressions and increases profibrotic signaling	Regulated by insulin and steroids	[71,72,73,74]
	**Adiponectin**	Encourages angiogenesis through the upregulation of Vascular endothelial growth factor (VEGF)-A, MMP-2 and MMP-9 Regulates glucose and lipid homeostasis and maintains insulin sensitivity	Reduced in obesity	Regulated by peroxisome-proliferator-activated receptor (PPAR)-γ ligands Inhibited by endoplasmic reticulum stress and proinflammatory cytokines	[75,76,77,78,79,80,81,82,83,84]
**Vasoconstrictor**	**Endothelin-1**	Endothelial cells release ET-1 in response to low shear stress and hypoxia to increase blood flow	ET-1 elevation in obesity resulting in vascular vasoconstriction and increased vascular permeability, adipocyte lipolysis, insulin resistance and endothelial inflammation, and perivascular fibrosis Can increase reactive oxidative species (ROS) production	Regulated by blood flow, leptin, and insulin	[85,86,87,88]
**Vasodilator**	**Nitric Oxide**	Adipose-derived nitric oxide inhibits lipolytic activity and promotes endothelial relaxation and vasodilation Adiponectin stimulates NO production	NO is reduced in obesity through TNF-mediated destabilization and free fatty acid (FFA) inhibition of NOS3 phosphorylation Increased ROS will quench NO	ROS FFA TNF	[89,90]
**Serpin (serine** **protease inhibitor)**	**PAI-1**	Induces fibrinogenesis by suppressing intravascular fibrinolysis	Marked increase in PAI-1 in individuals with obesity and diabetes (positive correlation with insulin resistance) When released into the blood stream, it negatively impacts vascular metabolism Fibrinolytic activities decrease in individuals with obesity individuals. Sustained impairment accelerates atherosclerosis	TNF TGF-β1 and Angiotensin II promote PAI-1 production in adipocytes and stromal cells Insulin responsive	[25,79,91,92,93]
**Growth Factors**	**VEGF**	Secreted by endothelial cells and adipocytes Acts as a chemotactic factor VEGF-A increases vascularization VEGF-B controls endothelial uptake of fatty acids	Suggestions of a positive VEGF correlation with BMI in humans Dysfunctional VEGF signaling results in impaired vascularization, increased vascular permeability, and endothelial dysfunction in individuals with obesity	SPARC regulates VEGF in individuals with diabetes Leptin modulates VEGF-A expression	[10,79,83,94,95,96,97,98]
	**PDGF**	Involved in angiogenesis and developmental adipogenesis ASCs become pro angiogenic in response to Platelet-derived growth factor (PDGF)	M1 macrophages overexpress PDGF-B during obesity PDGF-B induces proliferation and migration of aortic smooth muscle cells in vitro, which results in the thickening of the artery Myofibroblast mitogen that contributes to adipose fibrosis	Stromal-cell-derived factor 1 Positively regulated by IL-1β and TGF-β1	[99,100,101,102,103,104,105]
	**TGF-β**	Regulates the rate of adipogenesis	TGF-β1 and TGF-β3 increase and can cause mesenchymal transitions and dedifferentiation during obesity These isoforms increase basement membrane production, crosslinking, and inflammatory cytokine production in both adipocytes and endothelial cells	TNF can promote TGF-β secretion in differentiated adipocytes	[24,31,106,107]

### 2.2. Adipokine Secretion

There are several proteins produced by adipose tissue that function to regulate its energy balance. Leptin works through the central nervous system in order to regulate satiety [108]. This peptide hormone is primarily released from adipocytes, indicating that the amount in circulation has a positive correlation with the fat mass of the individual [109,110]. In obesity, where there is an elevated triacylglycerol level and an increased fat mass, leptin is elevated [72]. In healthy individuals, leptin works through the hypothalamus to reduce hunger, but in obesity, leptin resistance is not uncommon [111]. An individual with leptin resistance expresses a lack of satiety, resulting in further over-consumption and an increase in total body mass [112]. This hyperleptinemia that occurs in obesity has adverse cellular effects and has been implicated in the development and progression of atherosclerosis by stimulating the production and accumulation of ROS from endothelial cells [73,113,114].

It has been shown that many cells express the leptin receptor. Unsurprisingly, the paracrine effects of leptin are seen within the immune system, within the vasculature, and within the adipose lineage itself [115,116]. Importantly, leptin regulates the adipogenicity and insulin sensitivity of both adipose stem cells and adipocytes through Pre-B-cell leukemia homeobox (Pbx)-regulating protein-1 (PREP1). Notably, the administration of leptin on adipocytes and adipose stem cells reduced PREP1 while increasing TLR4 to limit the adipogenicity [116]. Leptin has been shown to have biphasic effects on preadipocyte and stromal vascular fraction (SVF) proliferation, with lower concentrations promoting proliferation but higher concentrations reducing proliferation [117].

It is known that insulin regulates the release of leptin from adipocytes; however, the effects remain inconsistent. When treated with 100 nM insulin, rat explants and rat-isolated adipocytes experienced a 30–50% reduction in leptin secretion, dependent upon co-treatment with steroids, dexamethasone or hydrocortisone, which elevated leptin secretion [118]. However, Wabitsch found a stimulatory dose-dependent response of insulin on leptin release from differentiated human adipocytes [71], and this was substantiated when 100 nM of insulin was administered to isolated adipocytes, resulting in a 10-fold increase in the secretion of leptin [119]. Further, Kolaczynski found that insulin did not stimulate leptin production acutely; however, over time, insulin increased the production of leptin [120,121].

There are contradictory results surrounding the relationship between leptin and VEGF. Morad et al. explored this relationship in the context of breast tissue and found that the exogenous inhibition of VEGF resulted in a decrease in adipocyte-derived leptin but that the inhibition of leptin did not alter VEGF secretion [122]. However, Nigro et al. showed that the administration of leptin to HUVECs drastically inhibited endothelial tube formation and cellular migration. This leptin administration was also shown to reduce VEGF-A, MMP2, and MMP9 protein levels [83]. This leptin-derived vascular dysregulation was corroborated by Cao et al. [123]. However, it was found that leptin induced angiogenesis and an increased vascular permeability through the upregulation of VEGF and fibroblast growth factor-2 (FGF-2) [123]. Further, obese adipose tissue has an increased secretion of VEGF that results in vascular smooth muscle proliferation, which contributes to vascular dysfunction [98]. However, this dysfunction has been shown to be ameliorated by the activity of adiponectin, an adipokine abundantly present in healthy individuals and lowered with obesity [97].

Adiponectin has been shown to have positive effects on adipocyte differentiation, promoting the more favorable hyperplastic growth rather than hypertrophic growth, and to assist in insulin sensitization [124,125]. In healthy individuals, adiponectin also has paracrine effects on both endothelial cells and macrophages. It has been demonstrated that adiponectin protects endothelial cells from the adverse effects of TNF through the inhibition of the NF-κB pathway [76]. For example, Kobashi et al. showed that adiponectin inhibits Interleukin-8 (IL-8) synthesis through the inhibition of the NF-κB pathway [126]. Additionally, adiponectin reduces the recruitment and restricts the lycollysis of infiltrating T cells, reducing obesity-induced adipose inflammation [63,127]. Individuals with obesity experience a reduction in adiponectin, allowing for a more proinflammatory and insulin-resistant environment [84]. While it is apparent that adiponectin has protective effects that contribute to insulin sensitization, the actions of adiponectin seem to be comparable to those of insulin in regard to its ability to encourage vasodilation and increase blood flow through its direct activation of endothelial nitric oxide synthase (eNOS), a regulator of vascular tone [128].

### 2.3. Vasoregulators in Obesity

Vascular tone is regulated by a series of vasodilators and vasoconstrictors. Under normal circumstances, these regulators work together to maintain homeostasis. However, in obesity, dysfunctional and inflamed adipose tissue secretes proinflammatory adipokines that impair normal vasoactivity, resulting in an imbalance in constrictors and dilators, including endothelin-1 and nitric oxide, respectively [85,89].

The vasoconstrictor, endothelin-1, is primarily secreted by endothelial cells and acts in both an autocrine and paracrine manner in adipose tissue, affecting both endothelial cells and adipocytes [129]. In obesity, ET-1 tends to be elevated, resulting in vasoconstriction, adipocyte lipolysis due to hormone sensitive lipase (HSL) phosphorylation, insulin resistance, and endothelial inflammation [130,131,132]. Similarly, reactive oxygen species are vasoconstrictors that exist as byproducts of aerobic metabolism [133]. It is well-founded that these entities, including hydrogen peroxide, hydroxyl radicals, and superoxide anions, induce cellular damage and inflammation [133,134]. The additional effects of ROS on the vasculature include a surge in intracellular Ca^2+^, which results in vascular contraction, increased permeability, and NO quenching, resulting in additional constriction [135]. Further, TNF activates the generation of ROS in HUVECs while also inducing the transcription of NF-κB and ERK ½ phosphorylation, which both upregulate ICAM-1 and VCAM-1 expressions on the endothelial surface to promote monocyte attachment, as previously mentioned [28,136,137,138].

Human adipose tissue expresses several enzymes responsible for the production of NO, a paracrine molecule whose primary function is to mediate endothelial relaxation, including membrane-bound eNOS and cytoplasmic inducible NO synthase (iNOS) [139]. It has been suggested that NO production in adipose tissue promotes the differentiation of adipose preadipocytes through the upregulation of PPARγ but that it also inhibits both basal- and catecholamine-stimulated lipolysis in subcutaneous tissue [140,141,142]. However, the bioavailability of NO is reduced during obesity. This occurs due to either TNF-mediated destabilization, the FFA-mediated inhibition of NOS3 phosphorylation, or through ROS-mediated quenching [89,90]. More explanation on this can be found in Sansbury et al.’s review on the subject [90]. As a consequence, this reduction in nitric oxide results in impaired vasodilation and tends to have pro-hypertensive effects [143].

Adiponectin has also been shown to be released from healthy perivascular adipose tissue and to travel trans-luminally in order to stimulate nitric oxide production and encourage the vasodilation of blood vessels [76,82,134]. However, this vasodilatory activity is inhibited in obese perivascular adipose tissue (PVAT), with low levels of adiponectin being correlated with impaired vasodilation [144]. Similar phenotypes were seen when an exogenous adiponectin blocker, a NO synthase inhibitor, TNF, and IL-6 were applied to healthy PVAT+vessels, indicating the role of inflammation and free radicals in PVAT-induced endothelial contractility [134]. This protective paracrine role of adiponectin on endothelial cells has been further shown to prevent the detrimental inflammatory activity of TNF on endothelial cells, as previously discussed [75,76].

## 3. Extracellular Matrix Remodeling during Obesity

Filamentous actin (f-actin) is the primary cytoskeletal filament that provides structural integrity within most cells. As an adipose progenitor cell matures into a unilocular adipocyte, there is a normal reduction in the filamentous actin to accommodate intracellular lipid storage. However, one characteristic of mice fed a high-fat diet was the massive increase in f-actin, actin polymerization machinery, and Rho Kinase activity in adipocytes that resulted in a reduction in insulin sensitivity, indicating that the mechanical stretch of the cell was transduced toward a pathological phenotype [13,145].

In order to accommodate cell size fluctuations, the tissue’s extracellular matrix (ECM) will dynamically remodel and allow for cell and tissue expansion, with healthy subcutaneous and omental adipose tissue expressing elevated levels of enzymes involved in ECM remodeling [146]. However, it has been shown that a consequence of obesity is a gradual accumulation of ECM. Henegar et al.’s transcriptomic analysis indicated that the chronic inflammation that occurs in obesity encourages macrophages to stimulate ECM production from neighboring pathological preadipocytes [147]. This ECM regulates the further expandability of the tissue, as well as nutrient diffusion to the cells, and future angiogenesis.

With the pericellular fibrosis of adipocytes and the vasculature being a consequence of obesity, Reggio et al. investigated both the composition and quantity of basement membrane proteins in the endothelial cells and adipocytes from lean, obese, and morbidly obese subcutaneous adipose tissue [106]. Here, it was found that both isolated endothelial cells and adipocytes express common basement membrane components, such as collagen IV and laminin; however, it was also found that adipocytes displayed more Heparan Sulfate Proteoglycan 2 (HSPG2/Perlecan) and Secreted Protein, Acidic, Rich in Cysteine (SPARC) than endothelial cells [106]. This suggests the existence of more crosslinks in the adipocyte matrix [148]. Further, TGF-β1 and TGF-β3 isoforms were upregulated alongside all basement membrane proteins in cellular fractions from individuals with obesity. The treatment of lean adipocytes and endothelial cells with these recombinant isoforms resulted in the upregulation of both inflammatory- and ECM-related genes, including, PAI-1, TGF-β1, connective tissue growth factor (CTGF), IL-6, collagen VI, SPARC, and lysyl oxidase (LOX). This indicates the involvement of both adipocytes and endothelial cells in the thickening of their respective basement membranes and releasing pro-fibrotic stimuli during obesity [106]. For further implications, please see our recent review on coordinated ECM remodeling, angiogenesis, and adipogenesis [149].

To corroborate the finding that basement membrane components derived from adipocytes and endothelial cells serve as pro-fibrotic stimuli, Kos et al. found positive correlations with SPARC levels and adipose fibrosis and metabolic dysfunction [150]. It was shown that leptin induces SPARC expression in adipose tissue and that SPARC expression was then correlated with IL-6 and fasting insulin levels [150]. SPARC has also been shown to further retinopathy in individuals with diabetes by interacting with VEGF and PAI-1 in endothelial cells [151]. In a 3D in vitro vasculature model, leptin was also shown to have profibrotic effects, indicated by the upregulation of collagens II and IV and the downregulation of MMPs 2 and 9 [28,80]. These pro-fibrotic effects were mitigated with the simultaneous addition of a counteracting adipokine, adiponectin (which is downregulated in obesity) [80]. This perivascular fibrosis has been shown to occur in individuals with obesity, where Spencer et al. showed that individuals with obesity have 58% less CD31-positive capillaries that were surrounded by more collagen V and less elastin [152]. Further, with the addition of collagen V to an HUVEC culture, angiogenesis was inhibited [152].

These matrix and paracrine alterations that occur in adipocytes and the vasculature during obesity can have severe implications if weight loss measures are not taken and the condition of low-grade inflammation is not resolved. Sasso et al. found an association between the adipose tissue elastography measurements on individuals with obesity taken prior to bariatric surgery and their degree of pericellular fibrosis, indicating the prominence of tissue stiffening and fibrosis in these same individuals [153]. These alterations also play a role in the development of obesity-induced comorbidities, including metabolic dysfunction and hypertension [154,155].

## 4. Obesity-Related Comorbidities: The Role of Adipose Tissue

It has been shown that the state of adipose tissue influences the behavior of endothelial cells, with obesity inducing endothelial dysfunction and interrupting the homeostatic paracrine signaling that occurs within healthy tissue [82]. It has also been indicated that endothelial cells from subjects with obesity tend to disrupt adipocyte function through the inflammatory signaling of IL-6 and IL-1β when co-cultured in vitro [156]. These obese endothelial cells reduced insulin sensitivity and lipolytic function of the adipocyte [156]. It is apparent that the crosstalk between cells in the obese state promotes a cycle of compounded inflammation that leads to other comorbidities, including cardiovascular disease and diabetes.

Adipose tissue has continuous capillaries with endothelial cells that form tight junctions and a persistent basement membrane in order to closely regulate the transport of molecules [157]. When dysfunctional, endothelial cells have an increased permeability and cause arterial stiffening. If endothelial dysfunction persists, there is a continuation of oxidative stress that leads to the elevation of ET-1, the reduction of NO, and the cycle of proinflammatory cytokines [89,158,159]. This impaired vasocrine signaling is characterized by the imbalance of vasoconstricting, and the released vasodilatory factors contribute to a proatherogenic and prothrombotic environment. The clinical implications of obesity-derived endothelial dysfunction include the development of atherosclerosis and hypertension.

Atherosclerotic lesions result from the deposition of fat into the artery walls, the infiltration of macrophages, and the successive development of macrophage foam cells. Notably, atherosclerosis may be accelerated by obese adipose tissue in both direct and indirect manners. It has been shown that the adipokine resistin happens to be elevated in obesity [160]. When applied to endothelial cells in vitro, this adipokine was observed to upregulate the expression of the monocyte chemoattractant chemokine, as well as monocyte adhesion proteins, while also increasing the production of ET-1 [161]. Additionally, adipocyte-derived inflammatory cytokines, including IL-6 and TNF, may have similar endothelial dysregulation effects that stimulate the development and progression of an atherosclerotic plaque [162]. In this regard, there is a direct pathological process that provokes inflammatory cell infiltration and lipid deposition [163,164,165]. Further, when an individual is obese, there is more tissue to supply blood to. Thus, the body attempts to increase blood flow in order to support the increased tissue mass, coinciding with arterial stiffening and chronic vasoconstriction and resulting in an elevated arterial blood pressure, also referred to as hypertension [143,166]. This condition can, in turn, result in exacerbated arterial stiffening, and, in conjunction with other obesity-related comorbidities, such as atherosclerosis, it can result in heart disease and/or thrombosis.

Moreover, the concomitant development of adipose fibrosis during obesity can result in detrimental metabolic consequences, including the deposition of ectopic lipids in other organs and insulin resistance [167,168,169,170]. Several review papers have delved into the current mechanistic understanding of the development and progression of adipose tissue fibrosis and its relationship with insulin resistance [152,169,171]. Briefly, DeBari et al. discussed the contribution of chronic hypoxia and inflammation to adipose fibrosis [169]. While hypoxia typically encourages angiogenesis as a normal homeostatic response, the vasculature fails to appropriately expand in obese adipose tissue, reducing capillary density and prolonging hypoxia. Further, pathological angiogenesis is a notable characteristic and consequence of insulin resistance. This insulin resistance is characteristic of Type 2 Diabetes, leading to elevated blood glucose, which furthers endothelial damage and impairs angiogenesis through ROS and advanced glycation end-product (AGE)-mediated complications [12,172,173]. The chronic damage to adipose tissue and its vascular supply results in a cyclical deterioration requiring urgent intervention.

## 5. Conclusions

Here, we lay out the current research into the paracrine-, autocrine-, vasocrine-, and matrix-dependent signaling that occurs between adipose tissue and its vasculature in the development and progression of obesity and its consequential diseases. Throughout the progression of obesity, there is an infiltration of inflammatory cells, including monocytes, macrophages, and T cells, that then drive a cycle of local and systemic inflammation [18]. Upon infiltration, proinflammatory signals, including TNF, interleukins, and interferons, promote the transdifferentiation of both adipocytes and endothelial cells into mesenchymal, pro-fibrotic cells that then sustain local inflammation [24,31]. Additionally, the dysregulation of adipose-specific hormones, leptin and adiponectin, in obese adipose tissue has been shown to have atherogenic and fibrotic consequences by stimulating the production of ROS and pericellular collagen, respectively [73,74,174]. With cardiovascular disease and adipose fibrosis developing concomitantly with obesity, it is important to investigate these interactions between adipocytes, the vasculature, and their surrounding matrix. There are still many unknowns regarding this complex communication; for example, there is evidence to suggest an adipocyte-dependent signaling loop revolving around leptin and VEGF; however, the directionalities of autocrine and paracrine activities have yet to be fully elucidated. Further, it has yet to be determined how vascularization impacts adipose fibrosis and metabolic dysfunction in humans. It is imperative that future research expands on these relationships in order to identify adipose-tissue-related therapeutics.

## Figures and Tables

**Figure 1 cells-12-00407-f001:**
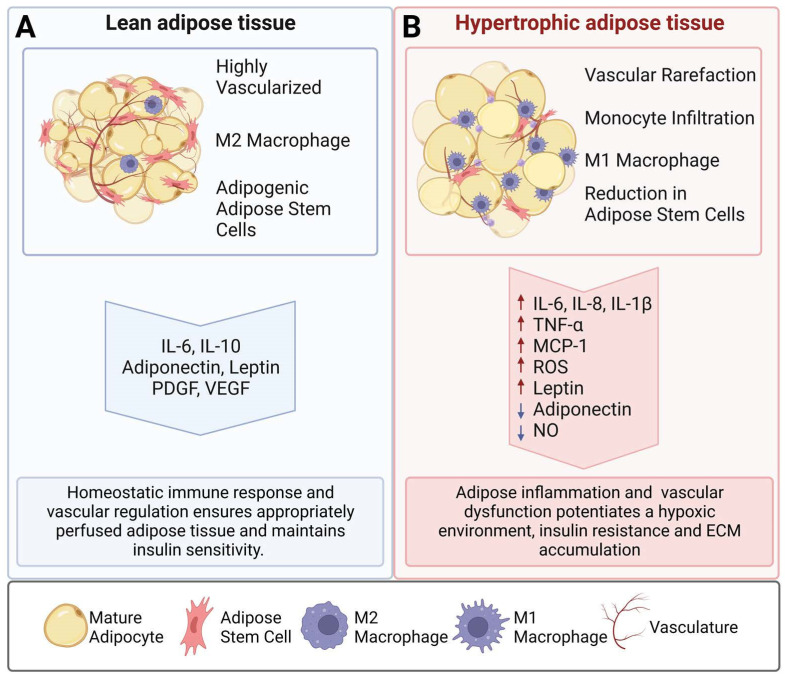
Cellular and functional differences in lean and obese adipose tissue. (**A**) Lean adipose tissue displays functional adipocytes that maintain a homeostatic balance of lipolysis and lipogenesis and is sufficiently vascularized to maintain the nutrient supply required for adipose tissue. (**B**) Obese adipose tissue is characterized by hypertrophic adipocytes that have a lower capillary density to supply the necessary nutrients, resulting in a hypoxic and proinflammatory environment.

**Figure 2 cells-12-00407-f002:**
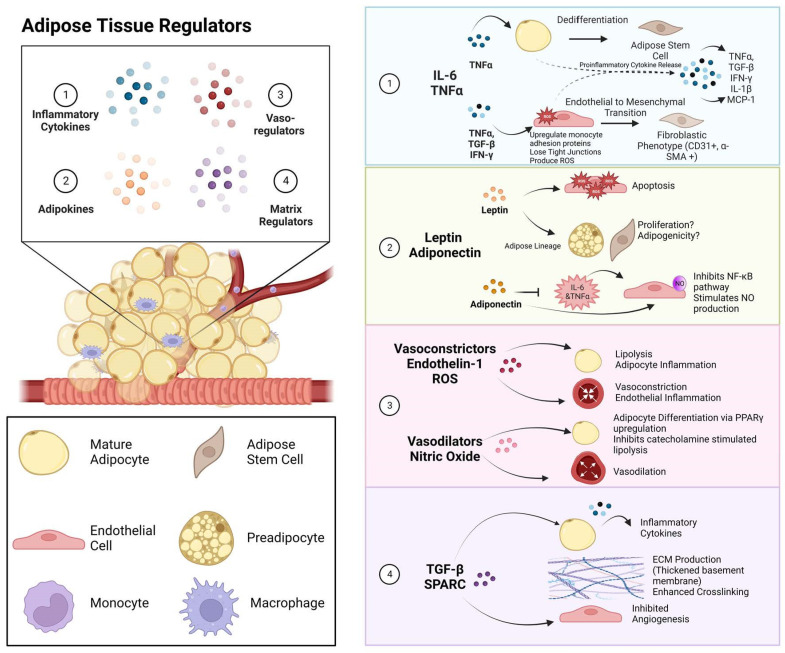
Paracrine, autocrine, and matrix signaling present in adipose tissue. Adipose tissue contains many subpopulations of cells, including inflammatory cells, stem cells and progenitors, and endothelial cells, which are in constant communication with each other and their surroundings. These signals occur in the following contexts: (1) Inflammatory cytokines, including IL-6 and TNF, tend to be abundantly present in obese adipose tissue. These cytokines stimulate a proinflammatory cascade in both adipocytes and the vasculature while also stimulating dedifferentiation within both adipocytes and endothelial cells. (2) Adipokines, leptin, and adiponectin are readily released from adipocytes but act in both an autocrine and paracrine manner, and leptin, a cytokine upregulated in obese adipose tissue, induces the production of reactive oxygen species in endothelial cells while affecting the proliferative capacity and adipogenicity of adipose precursor cells. (3) Vasocrine factors, including vasoconstrictors and vasodilators, are released by both adipocytes and endothelial cells to regulate vascular tone while also altering adipocyte functionality. (4) Matrix-regulating molecules are released by adipocytes and endothelial cells to alter the microenvironment of adipose tissue.

## Data Availability

Not Applicable.

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
