# Peer review of "Adipose Tissue Paracrine-, Autocrine-, and Matrix-Dependent Signaling during the Development and Progression of Obesity"

_cells, 2023, doi:10.3390/cells12030407_

Round 1

Reviewer 1 Report

This is well written, highly informative paper summarizing recent advances in our understanding of the adipose tissue paracrine, autocrine, and matrix-dependent signaling during obesity. The reviewer only has a few minor suggestions and comments as listed below.  

1. It is better to include keys for different cells in Figure 1. 

2. The authors want to confirm whether Ref 19 is properly cited in Line 70 as Ref 19 is about fibrosis, not matching the statement here.  

3. The left part of Figure 2 is not described in the figure legend. Besides, as illustrated in the right part of Figure 2, both autocrine and paracrine signalings are involved. Please consider revising the text content in Figure 2 and its legend to fit the statement better.  

4. Ref 22 in Line 84 is not properly cited, please consider removing or replacing it.  

5. Ref 34- 37 are from the 1990s. The readers will appreciate it if the authors want to cite some recent publications about the most updated progress in this topic. 

6. Not sure what the “adipose targeted IL-6” means at Line 144. 

7. Ref 57 at Line 146 and Ref 59 at Line 153 are review papers. Please cite the original research papers instead.    

8. References are needed for the statements in Lines 176-177 as well as Lines 265-266.  

9. Since this is a highly informative paper with lots of information to digest, to assist readers, it would be nicer if some of the expressions of the gene functions are more specific. For example, at Line 276, how does the ECM regulate further expandability, in a good or bad way? At Line 281, what are the findings of Reggio’s research Ref 108?  

10. As concluded in the abstract, "this review focuses on the signaling between adipocyte and endothelial cells",  thus the function of adipose tissue-derived VEGF in obesity could not be missed and deserves considerable discussion. The authors wish to add some paragraphs on this topic as well.  

Author Response

This is well written, highly informative paper summarizing recent advances in our understanding of the adipose tissue paracrine, autocrine, and matrix-dependent signaling during obesity. The reviewer only has a few minor suggestions and comments as listed below.  

  1. It is better to include keys for different cells in Figure 1. 

Thank you for your suggestion! A key has been added to Figure 1.

  1. The authors want to confirm whether Ref 19 is properly cited in Line 70 as Ref 19 is about fibrosis, not matching the statement here.

Thank you for catching this! This reference was removed.   

  1. The left part of Figure 2 is not described in the figure legend. Besides, as illustrated in the right part of Figure 2, both autocrine and paracrine signalings are involved. Please consider revising the text content in Figure 2 and its legend to fit the statement better.  

The caption for Figure 2 was revised in order to better capture both sides of the figure. Thank you for your suggestion!

  1. Ref 22 in Line 84 is not properly cited, please consider removing or replacing it.  

Thank you for catching this, this reference was removed.

  1. Ref 34- 37 are from the 1990s. The readers will appreciate it if the authors want to cite some recent publications about the most updated progress in this topic. 

Great suggestion! I have updated the references with recent research articles concerning insulin resistance!

  1. Not sure what the “adipose targeted IL-6” means at Line 144. 

This was changed to adipocyte-targeted Il-6 for better clarification.

  1. Ref 57 at Line 146 and Ref 59 at Line 153 are review papers. Please cite the original research papers instead.  

Thank you for catching this, this has been updated with the original research articles.

  1. References are needed for the statements in Lines 176-177 as well as Lines 265-266. 

Citations have been added to these lines! 

  1. Since this is a highly informative paper with lots of information to digest, to assist readers, it would be nicer if some of the expressions of the gene functions are more specific. For example, at Line 276, how does the ECM regulate further expandability, in a good or bad way? At Line 281, what are the findings of Reggio’s research Ref 108?  

Thank you for this suggestion. Clarification and explanation has been added between lines 312 and 322.

  1. As concluded in the abstract, "this review focuses on the signaling between adipocyte and endothelial cells",  thus the function of adipose tissue-derived VEGF in obesity could not be missed and deserves considerable discussion. The authors wish to add some paragraphs on this topic as well.  

            This was a wonderful suggestion. A paragraph on VEGF has been added between lines 227 and 242. Additionally, more information on VEGF has been added to the table.

Reviewer 2 Report

This review article is well-written, well organized, and suitable for publication in "Cells". 

The paper focuses on the paracrine, autocrine, and matrix-dependent signaling between adipocytes and endothelial cells in the context of the development and progression of obesity and related metabolic abnormalities. 

The topic is unique and pertinent to the field. The report summarizes the present understanding of communication between adipocytes and endothelial cells in the setting of obesity development and raises critical questions that may aid future research.

The authors adopt a unique approach to summarize the research on this specific issue and provide some innovative insights for future research. 

Everything is in order.

The conclusions are consistent with the evidence and arguments presented and they address the main question posed.

The references are appropriate.

Author Response

Thank you for all of your positive reviews!

Reviewer 3 Report

The submitted review entitled ‘Adipose Tissue Paracrine, Autocrine, and Matrix-dependent Signaling During the Development and Progression of Obesity’ written by Elizabeth K. Johnston and Rosalyn D. Abbott describes the mutual signaling of adipose tissue and endothelium in the context of the secretion of bioactive compounds in the development and progression of obesity. It is presented in four structured chapters including introduction, extensive characterization of paracrine signals in obesity, and accompanying processes of extracellular matrix remodeling, and the role of adipose tissue in obesity-related pathologies, than summarized in very moderate conclusions.

Obesity became a worldwide pandemic and the growing number of papers in this field indicate the need for such summaries. In my opinion, Authors covered the topic quite selectively, rather than in a comprehensive overview (there are already published reviews that address this topic, i.e. Li M et al. Front Cardiovasc Med. 2021;8:681581, doi:10.3389/fcvm.2021.681581; Sabaratnam R et al. Front Endocrinol. 2021;12:681290, doi:10.3389/fendo.2021.681290), creating a kind of signpost in a broad field of obesity research. The manuscript is very solid and well written with clear figures based on recent research articles and reviews. However, it requires revision and I suggest some modifications below.

My main comment relates to the conclusion. After reading the interesting story contained in the main part of the work, I felt a bit abandoned without any take-home message. I recommend to add the closing chapter describing future directions in the relationship between adipose tissue and the endothelium in context of the development of new therapeutic approaches against obesity-related pathologies and is thus of great interest in our increasingly obese society.

Also, in subchapter 2.1. the role of IFNγ was marginally treated.

The review has been prepared with good diligence, but I found some errors that do not diminish the reception of work:

- Keywords are missing

- Lines 9 and 29: Reference 1 refers to studies performed in 2017-2018, not 2016.

- lines 26, 39 and herein: The nomenclature of lipids provided by IUPAC (Biochemical Nomenclature and Related Documents, 2nd edition, Portland Press, 1992. Edited by C Liébecq., ISBN 1-85578-005-4) stated that tern ‘triglycerides’ should be replaced by ‘triacylglycerols’ to avoid interpretation that does not convey the intended meaning. ‘triglycerides’, taken literally, indicates three glycerol residues.

- The names of the chemical compounds are not necessarily provided in capital letters.

- line 71 and herein: According to the HUGO Gene Nomenclature Committee database, tumor necrosis factor alpha is now simply called tumor necrosis factor (TNF).

- lines 149 and 238: Please unify abbreviations

- line 248: The FFA abbreviation was not explained in the text.

- Line 290: Collagen type names are provided in Latin numerals.

Moreover, I highly recommend to attach the list of all abbreviations in the manuscript in the extra section that undoubtedly facilitate reading.

In summary, I listed some shortcomings of the submitted review. However, the work is well outlined and organized. Due to all of the above-highlighted aspects, the manuscript in the present form can be recommended for publication in Cells after minor improvements.

Author Response

The submitted review entitled ‘Adipose Tissue Paracrine, Autocrine, and Matrix-dependent Signaling During the Development and Progression of Obesity’ written by Elizabeth K. Johnston and Rosalyn D. Abbott describes the mutual signaling of adipose tissue and endothelium in the context of the secretion of bioactive compounds in the development and progression of obesity. It is presented in four structured chapters including introduction, extensive characterization of paracrine signals in obesity, and accompanying processes of extracellular matrix remodeling, and the role of adipose tissue in obesity-related pathologies, than summarized in very moderate conclusions.

Obesity became a worldwide pandemic and the growing number of papers in this field indicate the need for such summaries. In my opinion, Authors covered the topic quite selectively, rather than in a comprehensive overview (there are already published reviews that address this topic, i.e. Li M et al. Front Cardiovasc Med. 2021;8:681581, doi:10.3389/fcvm.2021.681581; Sabaratnam R et al. Front Endocrinol. 2021;12:681290, doi:10.3389/fendo.2021.681290), creating a kind of signpost in a broad field of obesity research. The manuscript is very solid and well written with clear figures based on recent research articles and reviews. However, it requires revision and I suggest some modifications below.

My main comment relates to the conclusion. After reading the interesting story contained in the main part of the work, I felt a bit abandoned without any take-home message. I recommend to add the closing chapter describing future directions in the relationship between adipose tissue and the endothelium in context of the development of new therapeutic approaches against obesity-related pathologies and is thus of great interest in our increasingly obese society.

Thank you for your suggestion! The conclusions have been updated with future directions and take away messages!

Also, in subchapter 2.1. the role of IFNγ was marginally treated.

IFNy was added to the table thank your for your attention to this!

The review has been prepared with good diligence, but I found some errors that do not diminish the reception of work:

- Keywords are missing

Thank you very much for catching this! Keywords have been added!

- Lines 9 and 29: Reference 1 refers to studies performed in 2017-2018, not 2016.

Thank you for your attention to this! This has been updated in line 29.

- lines 26, 39 and herein: The nomenclature of lipids provided by IUPAC (Biochemical Nomenclature and Related Documents, 2nd edition, Portland Press, 1992. Edited by C Liébecq., ISBN 1-85578-005-4) stated that tern ‘triglycerides’ should be replaced by ‘triacylglycerols’ to avoid interpretation that does not convey the intended meaning. ‘triglycerides’, taken literally, indicates three glycerol residues.

Excellent suggestion. Triglyceride has been replaced with triacylglycerol.

- The names of the chemical compounds are not necessarily provided in capital letters.

These have all been double checked and updated accordingly. Thank you!

- line 71 and herein: According to the HUGO Gene Nomenclature Committee database, tumor necrosis factor alpha is now simply called tumor necrosis factor (TNF).

Thank you for that information. All mentions of  TNF-α have been changed to TNF.

- lines 149 and 238: Please unify abbreviations

ERK ½ has been fixed in these lines. Thank you for catching this!

- line 248: The FFA abbreviation was not explained in the text.

Thank you for catching this! This abbreviation was added.

- Line 290: Collagen type names are provided in Latin numerals.

Thank you for noticing this! This was changed into latin numerals.

Moreover, I highly recommend to attach the list of all abbreviations in the manuscript in the extra section that undoubtedly facilitate reading.

Wonderful suggestion! A list of abbreviations has been added for this resubmission.

In summary, I listed some shortcomings of the submitted review. However, the work is well outlined and organized. Due to all of the above-highlighted aspects, the manuscript in the present form can be recommended for publication in Cells after minor improvements.